# Predictive Factors and Risk Assessment for Hospitalization in Chest Pain Patients Admitted to the Emergency Department

**DOI:** 10.3390/diagnostics14232733

**Published:** 2024-12-05

**Authors:** Nadya Kagansky, David Mazor, Ayashi Wajdi, Yulia Maler Yaron, Miya Sharfman, Tomer Ziv Baran, Dana Kagansky, Gal Pachys, Yochai Levy, Daniel Trotzky

**Affiliations:** 1Faculty of Medicine, Tel Aviv University, Tel Aviv 6997801, Israel; batiakag@gmail.com (N.K.); traumazor@gmail.com (D.M.); wajdi.ayashi@hotmail.com (A.W.); yulia2615@gmail.com (Y.M.Y.); dusiakag90@gmail.com (D.K.); gal.pachys1@moh.gov.il (G.P.); yochai.levy@moh.gov.il (Y.L.); danielt@shamir.gov.il (D.T.); 2Shmuel Harofeh Geriatric Medical Center, Be’er Ya’akov 7033001, Israel; 3Yitzhak Shamir Medical Center, Zerifin 1213500, Israel; 4Sheba Tel-Hasomer Medical Center, Ramat-Gan 5262000, Israel; 5School of Public Health, Faculty of Medical & Health Sciences, Tel Aviv University, Tel Aviv 6997801, Israel; zivtome@tauex.tau.ac.il

**Keywords:** chest pain, emergency department, acute coronary syndrome, hospitalization, risk stratification

## Abstract

Background: Chest pain is one of the most common reasons for emergency department (ED) visits. Patients presenting with inconclusive symptoms complicate the diagnostic process and add to the burden upon the ED. This study aimed to determine factors possibly influencing ED decisions on hospitalization versus discharge for patients with the diagnosis of chest pain. Methods: In the cohort study including 400 patients admitted to the emergency unit with a working diagnosis of chest pain, data on demographics, medical history, symptoms, lab results, and risk scores were collected from the medical records of patients admitted to the ED with a working diagnosis of chest pain. To reduce potential bias, the analysis was restricted to 330 patients who were referred to the ED by a primary care provider or clinic for chest pain. Results: Of 330 patients admitted to the ED, 58.5% were discharged, and 41.5% were hospitalized. Hospitalized patients were significantly older, with a median age of 70 versus 57 years for those discharged (*p* < 0.001). A higher proportion of hospitalizations occurred during the late-night shift. Significant predictors of hospitalization included hyperlipidemia (OR 3.246), diaphoresis (OR 8.525), dyspnea (OR 2.897), and hypertension (OR 1.959). Nursing home residents had a lower risk of hospitalization (OR 0.381). The area under the ROC curve for this model was 0.801 (95% CI: 0.753–0.848), indicating the predictive accuracy of the model in estimating the probability of admission. The HEART (history, ECG, age, risk factors, and troponin level) score was more effective than the TIMI (Thrombolysis in Myocardial Infarction) score in predicting the need for hospitalization, with an area under the curve (AUC) of 0.807 compared to 0.742 for TIMI. Conclusions: The HEART score in comparison with TIMI score proved especially valuable for quick risk assessment for hospitalization. The model that included hyperlipidemia, diaphoresis, dyspnea, and hypertension was the most predictive for the risk of hospitalization. Further research with larger populations is needed to validate these findings.

## 1. Introduction

Chest pain is the second most common complaint that drives patients to visit the emergency department (ED) and can be a symptom of many conditions from life-threatening heart problems to benign mild conditions. Less than half of patients admitted to the ED are diagnosed with acute coronary syndrome (ACS) [1,2], a life-threatening condition that requires immediate attention and precise diagnosis [3]. Patients with inconclusive clinical signs and symptoms require a more complex diagnostic process including ECG, blood tests for biomarkers, and sometimes more advanced tests. The fear of misdiagnosing a life-threatening condition occasionally leads to unnecessary referrals from primary physicians and adds to the ED burden [4]. The situation is even more complex in older patients as the presentation of ACS may involve different clinical variables and a higher ACS-related mortality rate than in younger adult patients [5,6,7].

Half of the patients arriving at the ED with chest pain do not fulfill the criteria of ACS (atypical symptoms, normal cardiac markers, no signs of ischemia on ECG), and their pain is categorized as Non-Cardiac Chest Pain (NCCP) or non-specific chest pain (NSCP) [2,6]. Even though many clinical trials focus on diagnosing and classifying NCCP, there is still a critical gap in clear and effective management and treatment strategies for these patients [7,8]. Several scores and tests have been developed and employed to distinguish between NCCP and ACS, such as the TIMI and HEART score, D-dimer, brain natriuretic peptide, electrocardiogram (ECG), echocardiography, cardiac computed tomography (CT), myocardial perfusion imaging, cardiac magnetic resonance imaging, and coronary angiography (CAG) [9,10,11]. Emerging evidence suggests that inflammatory markers such as the neutrophil-to-lymphocyte ratio (NLR) and platelet-to-lymphocyte ratio (PLR) may play a role in evaluating patients presenting with chest pain, but they have yet to be sufficiently validated and implemented in risk assessments [12,13,14]. However, even these examinations may not provide sufficient specificity for the diagnosis of NCCP [15,16]. This gap is crucial, given the significant impact of NCCP on patients’ quality of life, with the diagnosis often leading to psychological distress, functional limitations, and frequent healthcare visits [17,18], especially in older patients with cognitive problems and multiple comorbidities [19].

In this study, we conducted a comprehensive assessment of patients presenting to the emergency department with chest pain. We aimed to compare the characteristics of discharged and hospitalized patients to achieve a more focused assessment in the ED.

## 2. Materials and Methods

### 2.1. Study Population and Design

This study was performed at Israel’s fourth largest government hospital, Shamir (Assaf Harofeh) Medical Center, an 848-bed academic medical facility that provides care for over 1 million residents of Israel’s central region. The emergency medicine department is the fourth largest in Israel, treating about 160,000 patients each year.

The data for this cohort study were collected from computerized medical records spanning January 2019 to December 2019. All data collection procedures were conducted in accordance with the guidelines of the ethics (Helsinki) committee.

This time frame was chosen to ensure that the analysis focused on data unaffected by the COVID-19 pandemic. The pre-pandemic period was selected to isolate variables and outcomes from potential confounders such as changes in healthcare utilization, shifts in patient behavior, and disruptions to routine medical services caused by the pandemic. By excluding the pandemic period, this study aims to provide a more accurate and reliable assessment of the studied parameters under normal healthcare conditions.

In this cohort study, 400 patients admitted to the emergency unit with a working diagnosis of chest pain were included. Inclusion criteria were an age of 18 years and older with a diagnosis of chest pain upon admission to the ED. To reduce potential bias, the analysis was restricted to patients who were referred to the ED by a primary care provider or clinic for chest pain. No additional exclusion criteria were applied. This decision was made because patients in Israel have the option to visit the ED directly or with a referral from a primary physician. The choice between these options can depend on factors such as the distance between the patient’s location and the ED or clinic, the clinic’s operating hours, and the involvement of family members in the decision-making process. However, the majority of patients arrive with a referral, which tends to decrease the number of cases with non-cardiac symptoms. Therefore, to ensure consistency and focus on patients with a likely cardiac diagnosis, only those with referral letters were included in the statistical analysis.

### 2.2. Physician Groups and Training Protocols

The Emergency Department (ED) at Shamir Medical Center employs two distinct groups of physicians: senior physicians specializing in emergency medicine and physicians in fellowship programs for internal medicine or emergency medicine. The latter group includes physicians who have completed at least one year of work experience either in the ED or in an internal medicine department, in addition to having passed a specialized training program in the ED under the supervision of the Medical Director. Only Senior Physicians or experienced fellows nearing the end of their fellowship are authorized to make final decisions regarding patient admission or discharge. Approximately five such experienced physicians who completed the same training and adhered to consistent decision-making protocols were involved throughout the one-year study period. These physicians worked across all shifts (day, evening, and night) ensuring uniformity in patient care and decision-making processes.

### 2.3. Data Collection

The data were obtained from electronic medical records (EMRs) and included comprehensive patient information within the ED. Demographic details (age, gender, marital status, primary residence, functional status) and admission specifics (time, day, vital signs) were recorded. Clinical characteristics (chronic illnesses, symptoms) and laboratory findings (blood tests, cardiac markers, imaging) were also documented. ED diagnoses were categorized, and treatment details, including time intervals and specific interventions, were noted. Scoring systems such as TIMI and HEART were utilized for risk assessment. Study endpoints focused on hospitalization or discharge to community care from the ED.

### 2.4. ECG Changes

The ECG data were divided into four groups: 1—without ischemic changes; 2—with ischemic changes; 3—not specific changes; 4—with dysrhythmias without ischemic changes. A significant majority of these dysrhythmias were identified as supraventricular tachycardias (SVT). Some patients received treatment in the department prior to discharge. Due to the limited number of ventricular tachycardia (VT) cases (3), VT was not analyzed as a separate category.

### 2.5. Diagnosis at Admission

After investigation in the ED, various diagnoses of chest pain were categorized into 7 groups: abdominal cause, ACS, respiratory cause, trauma, arrhythmia, chest pain, and other causes.

The HEART and TIMI scores are validated risk scores for ACS in patients presenting with chest pain [11,12,13].

### 2.6. Statistical Methods

Categorical variables were summarized as frequencies and percentages. The distribution of the continuous variables was evaluated using histograms and Q-Q plots. Normally distributed continuous variables were presented as means and standard deviations (SDs), while other variables were presented as medians and interquartile ranges (IQRs). The Chi-square test and Fisher exact test were employed to compare categorical variables between admitted and non-admitted patients. The Mann–Whitney test and the independent samples T-test were used to compare the continuous variables between these groups. In the first model of multivariable analysis, variables that showed statistically significant associations with the outcome in univariate analysis were included. In the second model, clinically important variables that did not reach significance but have clinical significance were added. Logistic regression was utilized for the multivariable analysis. In both models, the backward stepwise method was applied, with a criterion for removal based on a *p*-value of >0.1 in the Wald test. The area under the receiver operating characteristic (ROC) curve was used to evaluate the discriminative ability of TIMI and HEART scores and the final model for distinguishing between admitted and non-admitted patients. The DeLong test was used to compare the areas under the ROC curves (AUCs) between the TIMI score and the HEART score. All statistical tests were two-sided, and a *p*-value of < 0.05 was considered statistically significant. Statistical analyses were performed using SPSS software (IBM SPSS STATISTIC for Windows, version 28, IBM Corp., Armonk, NY, USA, 2021).

## 3. Results

The records of 400 patients admitted to the emergency room of Shamir Medical Center with the diagnosis of chest pain between January 2019 to December 2019 were reviewed. Seventy of them arrived without a referral, leaving 330 patients in the cohort. Out of this cohort, 193 patients (58.5%) were discharged following ED investigation and treatment, while 137 patients (41.5%) required hospitalization for further evaluation. Within 7 days from discharge, there were three deaths among the hospitalized patients (2.2%) and two deaths among those discharged (1.0%). Regarding readmissions, within 24 h, five discharged patients (2.6%) were readmitted, followed by an additional three patients (1.6%) within 48 h and two more patients (1.0%) within 72 h. Given the small number of events, we present these data descriptively.

### 3.1. Comparison of Demographic Characteristics and Chronic Illnesses Between the Two Cohorts

Demographic data are presented in Table 1. In both groups, men comprised a slightly higher percentage (57.6%) than women (42.4%). Most patients were married, with single individuals representing a smaller group (15.5% discharged and 5.1% hospitalized). Single individuals demonstrated a higher discharge rate (*p* = 0.007). The discharged cohort patients were younger, with 58% of individuals being younger than 65 years, compared to only 24.1% in the hospitalized cohort. Conversely, the hospitalized group had higher proportions in the older age categories, particularly in the 65–74 (44.5% versus 26.4%) and in the 75–84 (23.4% versus 10.9%) age ranges. Patients aged 85 years and above accounted for 8% of hospitalizations and 4.7% of discharges. Nursing home residents were significantly more likely to be discharged compared to patients living at home (*p* < 0.001). Most patients were classified as having independent basic physical function and preserved cognition, with no difference between hospitalized and discharged patients. In the hospitalized group, there were higher percentages of risk factors for ischemic heart disease (IHD) [20] compared to the discharged cohort. Polypharmacy was observed in 24.9% of discharged patients compared to 57.4% of hospitalized patients (*p* < 0.001).

### 3.2. Clinical Symptoms and Laboratory Data

The clinical symptoms and the laboratory data revealed notable distinctions between the hospitalized and discharged patients (Table 2). Hospitalized patients exhibited a higher prevalence of symptoms such as fatigue (24.3% vs. 12.7%, *p* = 0.007), diaphoresis (14.7% vs. 2.1%, *p* < 0.001), and dyspnea (35.3% vs. 15.4%, *p* < 0.001). There was no significant difference in symptoms like palpitations, dizziness, and nausea. The pain characteristic data further emphasized distinctions between these groups. Hospitalized patients were less likely to have missing pain duration data and more likely to report pain the days before admission, radiating pain, and chest pain compared to discharged patients. Notably, there were no significant differences in reports of pain hours before admission or pain fluctuation between the two groups. Ischemic ECG changes were more prevalent among hospitalized patients. Other dysrhythmias without ischemic changes were slightly more common in the hospitalized group.

Significant variations in median hemoglobin levels, median potassium levels, estimated glomerular filtration rate (eGFR), C-reactive protein (CRP), and troponin levels were observed between the two groups (Table 3). Interestingly, no significant differences were noted in white blood cell count (WBC), creatine phosphokinase (CPK), D-dimer, and creatinine (CRE) levels. NLR and PLR were higher in admitted patients.

### 3.3. Diagnosis Classifications and Treatment at the ED

A significantly higher proportion of patients with ACS were hospitalized (12.4% vs. 0.5%, *p* < 0.001), while a significantly higher proportion of patients with abdominal causes were discharged (2.2% vs. 8.3%, *p* < 0.001), as shown in Table 4. Other diagnosis classifications, including respiratory, trauma, arrhythmia, chest pain, and others, did not show notable differences between the two groups. Hospitalized patients experienced shorter median times from reception to triage (6 min) and from reception to first medical doctor (MD) examination (54 min) compared to discharged patients (14.5 min and 82.5 min, respectively) (*p* < 0.001 for both). Hospitalized patients had significantly higher rates of oxygen treatment and fluid infusions than discharged patients. During the late nurse night shift (23:00–7:00), more patients were hospitalized (62%) than discharged (48.2%). In the evening shift (15:00–23:00) and the day shift (07:00–15:00), higher percentages of patients were discharged (36.3% and 15.5%, respectively) compared to hospitalized (27.7% and 10.2%, respectively, *p* = 0.042). Across weekdays, there were no notable differences in patient distribution between discharged and hospitalized groups. Within 24, 48, and 72 h, only 3.6% of discharged patients were readmitted to the ED, with all readmitted patients subsequently discharged (*p* = 0.044). There were no significant differences in death rates between hospitalized and discharged patients within 7 days from discharge (1.5% vs. 0.9%, *p* = 0.572).

### 3.4. TIMI and HEART Scores

Among discharged patients, 49.7% had a TIMI score of 0, compared to only 10.2% of hospitalized patients (Table 5). Conversely, 75.9% of hospitalized patients had a TIMI score of 1–3, while 47.2% of discharged patients fell into this category. Higher TIMI scores (4+) were more prevalent among hospitalized patients (13.9%) compared to discharged patients (7.6%) (*p* < 0.001). The distribution of HEART scores demonstrated that discharged patients were more likely to have lower HEART scores (3 or below) compared to hospitalized patients (69.9% vs. 22.6%, *p* < 0.001). In contrast, hospitalized patients had higher HEART scores (4–6 and 7+) compared to discharged patients (62% vs. 27.5% for 4–6; 15.3% vs. 2.6% for 7+, both *p* < 0.001).

The ROC curve (Figure 1) assesses the predictive ability of the TIMI and HEART scores for hospitalization. The TIMI score was 0.742 (95% CI: 0.689–0.796), and the HEART score was 0.807 (95% CI: 0.761–0.853). Both scores were statistically significant (*p* < 0.001), with the HEART score showing a higher predictive value compared to the TIMI score (TIMI-HEART, *p* < 0.001).

The discrimination ability of NLR and PLR was relatively low (AUC (area under the curve) 0.64 for NLR and 0.63 for PLR).

### 3.5. Factors Associated with Discharge/Hospitalization

The logistic regression analysis including factors that were significant in the univariate analysis (Table 6) revealed that living in a nursing home (OR 0.387, *p* = 0.006) and admission to the ED between 15:00 and 23:00 (OR 0.494, *p* = 0.032) were associated with reduced risk for hospitalization compared to arrival between 07:00 and 15:00. Conversely, hyperlipidemia (OR 3.514, *p* < 0.001), specific ECG changes (ECG changes 2 compared to ECG changes 1: OR 7.992, *p* = 0.013), diaphoresis (OR 7.043, *p* = 0.003), dyspnea (OR 3.641, *p* < 0.001), and elevated troponin levels (OR 3.820, *p* = 0.019) were associated with an increased risk of hospitalization.

In the expanded regression model, clinical and laboratory data were organized to identify predictive factors for hospitalization, including factors that were significant in the univariate analysis along with supposedly significant clinical factors that did not show significance in the univariate analysis such as age group, gender (female), primary residence, nurse shifts, IHD, CRF, s/p CVA/TIA, arrhythmia, hyperlipemia, hypertension, diabetes mellitus, polypharmacy, fatigue, diaphoresis, and dyspnea (Appendix A). The regression identified that living in a nursing home was significantly linked with lower odds of hospitalization (OR 0.381, *p* = 0.003), while hyperlipemia (OR 3.246, *p* < 0.001), diaphoresis (OR 8.525, *p* < 0.001), dyspnea (OR 2.897, *p* < 0.001), and hypertension (OR 1.959, *p* = 0.033) were significant risk factors for hospitalization. Fatigue (OR 1.885, *p* = 0.069) did not show a significant association with hospitalization (Table 7).

The area under the ROC curve for this model (Figure 2) was 0.801 (95% CI: 0.753–0.848), *p* < 0.001, indicating the predictive ability of the model in estimating the probability of admission.

## 4. Discussion

This study aimed to identify predictive factors and assess the risk of hospitalization for patients presenting with chest pain in the emergency department (ED). By analyzing demographic data, clinical symptoms, laboratory findings, and risk scores, this study sought to differentiate between patients who required hospitalization and those who could be safely discharged. Additionally, this study evaluated the predictive value of established scoring systems, specifically the HEART and TIMI scores, to enhance clinical decision-making and optimize resource allocation in ED settings.

This cohort study compared 330 patients who were referred to a tertiary hospital ED with a diagnosis of chest pain. Following evaluation in the ED, 193 (58.5%) were discharged, while 137 (41.5%) required hospitalization. These findings are comparable with international studies reporting that approximately 40–50% of patients presenting with chest pain to the ED are discharged after initial evaluation [21,22,23]. Fast and effective decision-making is crucial for these patients. This study examined the key factors associated with hospitalization risk for patients presenting with chest pain. Age and marital status were found to potentially influence the hospitalization rate, with older patients being more likely to require hospitalization for chest pain. This may be a result of the more complex evaluation in older adults who often present with atypical symptoms [24,25]. The higher prevalence of cardiac risk factors probably also has an impact. Finally, the lack of sufficient education in geriatrics may also affect the decision-making in the ER. While hospitalization can offer elderly patients thorough care, potential drawbacks such as infections and decline in function and cognition must be taken into account [26,27]. Providing emergency care to older patients requires a delicate balance between prompt intervention and comprehensive evaluation while respecting their discharge preferences whenever possible [24,25].

Interestingly, our findings on nursing home residents’ reduced hospitalization rates resonate with recent studies on post-acute care. Templeton et al. demonstrated that specialized nursing facilities reduce hospital readmissions and mortality, supporting the potential of such care models to alleviate ED overcrowding and improve outcomes for elderly patients [28]. This finding underscores the importance of integrating community-based interventions into broader healthcare strategies.

The higher prevalence of IHD risk factors in hospitalized patients is expected, indicating good decision-making, and needs no further explanations [29,30]. Despite technological advancements, thorough anamnesis remains vital, as ACS can manifest with non-specific symptoms [31,32,33,34,35]. The identification of symptoms such as diaphoresis and dyspnea as significant predictors is consistent with findings by Canto et al., which highlighted these symptoms as red flags for ACS [31]. Dyspnea, in particular, has been linked to increased mortality and hospitalization in studies like that of Keijzers et al., emphasizing the need for heightened vigilance when this symptom is present [32]. In contrast to observations by Williams et al. that fatigue is a non-specific symptom with low diagnostic utility in chest pain evaluation [33], fatigue was linked to hospitalization in our study. Palpitations, dizziness, and nausea did not show a difference between discharged and hospitalized patients also, suggesting these symptoms may be less indicative of cardiac events or not perceived as red flags by physicians [34,35]. Additionally, patients reporting pain days before admission or with radiating/fluctuating pain were more frequently hospitalized, indicating that such pain characteristics are more likely associated with severe cardiac events necessitating inpatient care. Recognizing these patterns aids clinicians in making informed decisions about managing chest pain patients. NLR and PLR showed differences between the groups; however, the discrimination ability of NLR and PLR was relatively low. In this study, patients were admitted after an initial evaluation and referral to the emergency department. This delay in ED arrival may have influenced the NLR and PLR results. Further research is needed to evaluate the differences between patients admitted directly to the emergency department and those admitted following a referral, as well as to assess changes in NLR and PLR over time in acute settings.

This study also sheds light on the operational dynamics of the ED, with higher hospitalization rates during night shifts. This trend remains consistent throughout the various days of the week, indicating a uniform demand and patient profile in the ED regardless of the day. This pattern aligns with research by Gorski et al., which explored the impact of ED crowding and physician workload on admission rates [36]. Studies have found that increased ED crowding and physician workload during peak hours are associated with higher admission rates, highlighting how operational pressures can influence clinical decisions [36] potentially due to cognitive fatigue and accumulating work demands [37,38]. These findings underscore the need for risk stratification tools to optimize resource allocation in clinical practice. Notably, interns in Israel still work long shifts of up to 26 h. Expanding the research to evaluate if shorter shifts lead to better decision-making and fewer hospitalizations could provide valuable insights.

Furthermore, we observed that patients who are ultimately hospitalized undergo significantly faster processing from reception to initial medical assessment compared to those who are discharged. This expedited care points to good triage in the ER focusing on high-risk patients [39]. However, the overall decision-making time from reception to the final decision of patient disposition shows no significant differences between discharged and hospitalized patients, probably due to a more comprehensive evaluation process in hospitalized patients.

Non-ischemic changes or dysrhythmias without ischemic changes were more common in discharged patients. Such ECG changes as non-specific ST-T changes, left ventricular hypertrophy patterns, bundle branch blocks, and early repolarization are usually chronic changes and indicate a lower risk profile sometimes suitable for outpatient care [40]. Furthermore, many arrhythmias such as atrial fibrillation and flutter can be treated in the ER and do not necessitate hospitalization.

Our study shows the importance of the TIMI and HEART scores in differentiating between hospitalized and discharged patients, highlighting their varying risk profiles. Both scores were higher in hospitalized patients, indicating effectiveness in identifying high-risk patients [41]. The HEART score showed a superior predictive capability compared to the TIMI score in this study, emphasizing its utility in the ED setting. It is important to note that TIMI and HEART scores are well known and widely used; however, our findings suggest that while some clinicians may prefer using one score over the other, the HEART score was significantly better in our study in aiding difficult decisions regarding patient hospitalization, particularly in complex cases. The literature has suggested that the TIMI score is better for predicting major cardiac events in patients with unstable angina or non-ST-segment-elevation myocardial infarction [42], while the HEART score has been validated for rapid risk stratification of chest pain in the emergency setting [43]. Other studies have also shown that the HEART score outperforms both TIMI and GRACE scores in predicting major adverse cardiac events (MACEs); it offers accuracy and reliability in identifying low-risk patients suitable for discharge, making it a more effective tool for risk assessment in the ED. Our results support these findings and highlight the HEART score’s significant advantage in emergency settings [44,45].

In the logistic regression analysis, living in a nursing home was recognized as a contributing factor associated with a reduced risk of hospitalization. This relationship could stem from the comprehensive care and observation received in these facilities, leading to a reduced need for hospital admissions. Moreover, specialized nursing homes for post-acute care have demonstrated a decrease in death rates and hospital readmissions, showcasing the benefits of this specialized care in mitigating hospitalizations [28].

Conversely, admission during the night hours, the presence of hyperlipidemia, specific ECG changes, diaphoresis, dyspnea, and elevated troponin TIMI levels were associated with a higher risk of hospitalization. Hyperlipidemia is commonly associated with cardiovascular diseases, often requiring hospital care. This connection is reinforced by studies showing that cardiovascular risk factors, including hyperlipidemia and hypertension, significantly contribute to the probability of hospital admissions and adverse outcomes [46,47]. Diaphoresis can signal severe conditions such as myocardial infarction, warranting immediate medical attention [31]. Dyspnea is frequently linked to heart failure and serves as a major factor leading to frequent hospital admissions [32]. Using significant factors identified in the univariate analysis with clinically significant factors, we formulated a scale to evaluate the chance of hospitalization in chest pain patients admitted to the ED. After logistic regression analysis, the scale included hyperlipemia, diaphoresis, dyspnea, place of residence, hypertension, and fatigue emerging as the most influential. The ROC curve analysis indicates that this logistic regression model serves as a good predictor of hospitalization and is comparable to the TIMI and HEART scores.

In summary, this study identified major factors influencing the likelihood of hospitalization for patients presenting with chest pain to the ED. Chronic risk factors for ischemic heart disease, diaphoresis, and dyspnea, as well as ischemic ECG changes, are significant predictors of hospitalization. The HEART score demonstrated a higher predictive capability than the TIMI score, making it particularly valuable for rapid risk assessment in the ED. The insights gained from this study could help healthcare providers make informed decisions, improve patient outcomes, and optimize resource allocation in clinical practice. Further research across multiple centers with larger and more diverse populations is required to validate these findings.

## 5. Limitations

While this study provides important insights, limitations such as its single-center design and retrospective nature should be acknowledged. Future research should focus on validating these findings in multicenter studies with diverse populations. Additionally, further exploration of long-term outcomes related to discharge versus hospitalization decisions would provide a more comprehensive understanding of patient trajectories.

## 6. Conclusions

This study provides valuable insights into factors predicting hospitalization for patients presenting with chest pain in the emergency department (ED). By focusing on pre-pandemic data and employing robust statistical methods, we identify symptomatology, clinical history, and risk assessment tools as critical determinants of hospitalization. Advanced age, chronic risk factors for ischemic heart disease, specific symptoms such as diaphoresis and dyspnea, and ischemic ECG changes are significant predictors of hospitalization. The effective use of TIMI and HEART scores in risk stratification underscores their utility in clinical decision-making. Notably, the HEART score demonstrated a higher predictive capability than the TIMI score, making it particularly valuable for rapid risk assessment in the ED. The insights gained from this study could help healthcare providers make informed decisions, improve patient outcomes, and optimize resource allocation in clinical practice. Further research across multiple centers with larger, diverse populations is needed to validate these findings.

## Figures and Tables

**Figure 1 diagnostics-14-02733-f001:**
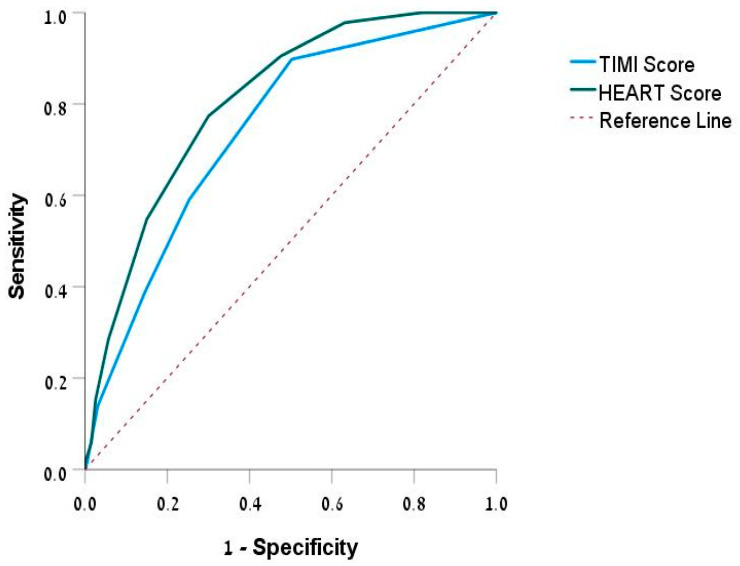
ROC curve of TIMI and HEART score. Abbreviations: ROC, receiver operating characteristic; TIMI, Thrombolysis in Myocardial Infarction; HEART, history, ECG, age, risk factors, troponin.

**Figure 2 diagnostics-14-02733-f002:**
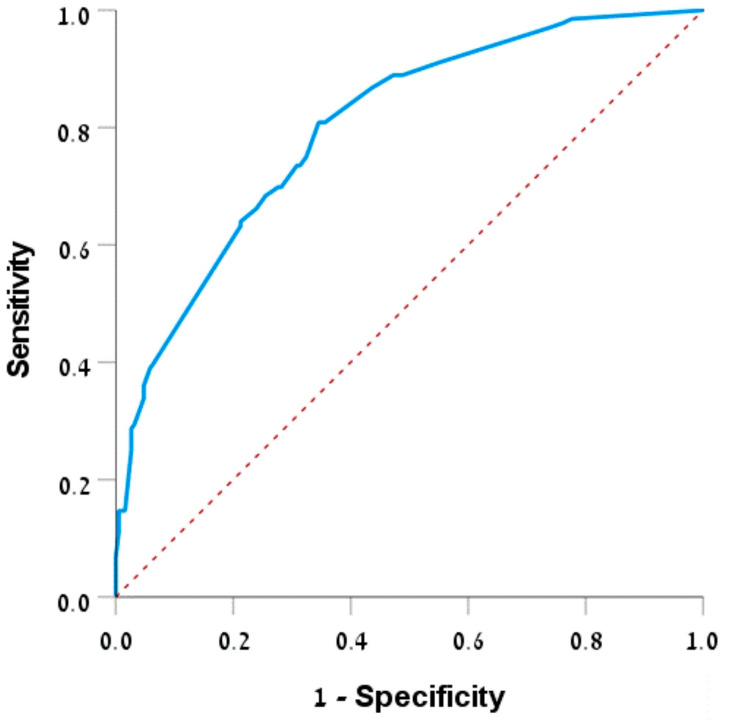
ROC curve—prediction of admission. Abbreviations: ROC, receiver operating characteristic.

**Table 1 diagnostics-14-02733-t001:** Basic data: demographic and chronic illnesses according to discharge versus hospitalization.

		Total		Discharged		Hospitalized		
		330	(%)	193	58.5%	137	41.5%	P
Gender	Men	190	57.6%	117	60.6%	73	53.3%	0.184
	Women	140	42.4%	76	39.4%	64	46.7%	
Marital status	married	236	71.7%	138	71.5%	98	72.1%	0.007
	widowed	29	8.8%	13	6.7%	16	11.8%	
	single	37	11.2%	30	15.5%	7	5.1%	
	divorced	27	8.2%	12	6.2%	15	11%	
Age group	Below 65	145	43.9%	112	58%	33	24.1%	<0.001
	Age 65–74	112	33.9%	51	26.4%	61	44.5%	
	Age 75–84	53	16.1%	21	10.9%	32	23.4%	
	Age 85+	20	6.1%	9	4.7%	11	8%	
Age (IQR)		66	46–73	57	38.5–70.5	70	65–76.5	<0.001
Primary residence	Home	227	68.8%	113	58.5%	114	83.2%	<0.001
	Nursing home	103	31.2%	80	41.5%	23	16.8%	
Basic physical function	dependent	16	4.8%	6	3.1%	10	7.3%	0.081
	independent	314	95.2%	187	96.9%	127	92.7%	
Basic cognitive function	preserved	326	98.8%	191	99%	135	98.5%	>0.999
Basic physical function	dependent	16	4.8%	6	3.1%	10	7.3%	0.081
IHD		99	30%	45	23.3%	54	39.4%	0.002
CRF		20	6.1%	5	2.6%	15	10.9%	0.002
Past MI		72	21.8%	30	15.5%	42	30.7%	0.001
Catheterization background		88	26.7%	37	19.2%	51	37.2%	<0.001
s/p CVA_TIA		17	5.2%	4	2.1%	13	9.5%	0.003
Arrhythmia		50	15.2%	21	10.9%	29	21.2%	0.01
Hyperlipidemia		149	45.2%	56	29%	93	67.9%	<0.001
Hypertension		170	51.5%	73	37.8%	97	70.8%	<0.001
Diabetes		104	31.5%	45	23.3%	59	43.1%	<0.001
Polypharmacy		126	38.3%	48	24.9%	78	57.4%	<0.001

Abbreviations: IHD, ischemic heart disease; CRF, chronic renal failure; MI, myocardial infarction; s/p CVA/TIA, status after cerebrovascular accident/transient ischemic attack; P, probability value; IQR, interquartile range.

**Table 2 diagnostics-14-02733-t002:** Clinical symptoms—according to discharge versus hospitalization.

		Total		Discharge		Hospitalization		
		N	(%)	N	(%)	N	(%)	P
Level of consciousness	Full consciousness	329	99.7%	193	100%	136	99.3%	0.415
	Moderately reduced	1	0.3%	0	0%	1	0.7%	
Condition at admission	Stable	328	99.4%	193	100%	135	98.5%	0.172
Fatigue		57	17.5%	24	12.7%	33	24.3%	0.007
Dizziness		19	5.8%	9	4.8%	10	7.4%	0.326
Diaphoresis		24	7.4%	4	2.1%	20	14.7%	<0.001
Confusion		1	0.3%	1	0.5%	0	0%	>0.999
Palpitations		31	9.5%	14	7.4%	17	12.5%	0.123
Dyspnea		77	23.8%	29	15.4%	48	35.3%	<0.001
Nausea		28	8.6%	15	7.9%	13	9.6%	0.607
Pain duration	No data	97	29.7%	73	38.4%	24	17.5%	<0.001
	Hours before admission	120	36.7%	67	35.3%	53	38.7%	
	Days before admission	85	26%	38	20%	47	34.3%	
	Fluctuating during period	23	7%	11	5.8%	12	8.8%	
Radiating pain		81	27.4%	39	21.5%	42	36.5%	0.005
Epigastric pain		40	12.3%	28	14.8%	12	8.8%	0.105
Chest pain		313	94.8%	176	91.2%	137	100%	<0.001
ECG changes	Without ischemic changes	229	75.1%	142	81.1%	87	66.9%	<0.001
	With ischemic changes	20	6.6%	2	1.1%	18	13.8%	
	Non-specific changes	20	6.6%	12	6.9%	8	6.2%	
	Other dysrhythmias without ischemic changes	36	11.8%	19	10.9%	17	13.1%	

Abbreviations: ECG, electrocardiogram; P, probability value.

**Table 3 diagnostics-14-02733-t003:** Laboratory data according to discharge versus hospitalization.

		Total		Discharge		Hospitalization		P
Total blood test		320	97.9%	184	96.8%	136	99.3%	0.246
WBC (IQR)	1000/µL	8.1	6.45–9.7	8.3	6.475–9.625	8	6.4–9.9	0.652
HB (IQR)	g/dL	13.6	12.575–14.7	13.9	12.9–14.925	13.4	11.925–14.2	<0.001
Na (IQR)	mmol/L	138	137–140	138	137–140	138	137–140	0.170
K (IQR)	mmol/L	4.1	3.9–4.4	4.1	3.8–4.3	4.2	3.9–4.6	0.004
CPK (IQR)	U/L	92	62–148.50	95	66–149	86	56.50–148.50	0.282
D Dimer (IQR	ng/mL	710	229.50–1632.50	229.50	157–1295.75	1362	503.50–4151	0.190
CRE (IQR)	mg/dL	0.86	0.73–0.99	0.86	0.71–0.96	0.865	0.74–1.025	0.114
eGFR (mean ± sd)	mL/min/1.73 m^2^	82.353 + 27.263		86.907 + 24.825		76.389 + 29.199		0.001
CRP (IQR)	mg/L	3.44	1.32–9.93	3.125	1.21–6.232	6.17	1.77–23	0.003
Troponin (IQR	ng/L	0.013	0.013–0.02	0.013	0.013–0.013	0.013	0.013–0.03	<0.001
Saturation (IQR)	%	98	97–99	98	97–100	98	96–99	0.033

Abbreviations: WBC, white blood count; HB, hemoglobin; Na, sodium; K, potassium; CPK, creatine phosphokinase; CRE, creatinine; eGFR, estimated glomerular filtration rate; CRP, C-reactive protein; P, probability value; IQR, interquartile range.

**Table 4 diagnostics-14-02733-t004:** Diagnosis and treatment at ED according to discharge versus hospitalization.

		Total		Discharge		Hospitalization		
		N	(%)	N	(%)	N	(%)	P
ED diagnosis classification	Abdominal cause	19	5.8%	16	8.3%	3	2.2%	<0.001
	ACS	18	5.5%	1	0.5%	17	12.4%	
	Respiratory	12	3.6%	5	2.6%	7	5.1%	
	Trauma	14	4.2%	11	5.7%	3	2.2%	
	Arrhythmia	11	3.3%	6	3.1%	5	3.6%	
	Chest pain	224	67.9%	134	69.4%	90	65.7%	
	Other	32	9.7%	20	10.4%	12	8.8%	
Time in minutes from reception to triage (IQR)		9	3–20.50	14.50	4.25–24.75	6	2–16.5	<0.001
Time in minutes from reception to first MD (IQR)		65.50	41–115	82.50	47–133.75	54	37–88	<0.001
Time in minutes from reception to decision (IQR)		234	172–318	224.50	161–317.25	241	183–321.50	0.159
Time in minutes from decision to hospitalization (IQR)		41	6.50–73.50	NA	NA	41	6.50–73.50	NA
Opiates		6	1.8%	1	0.5%	5	3.6%	0.086
Oxygen treatment		5	1.5%	0	0%	5	3.6%	0.012
Fluid infusion		16	4.9%	5	2.6%	11	8%	0.026
ED cardiologist consultant		43	13.1%	7	3.7%	36	26.3%	<0.001
Nurse shift	23:00–7:00	178	53.9%	93	48.2%	85	62%	0.042
	15:00–23:00	108	32.7%	70	36.3%	38	27.7%	
	07:00–15:00	44	13.3%	30	15.5%	14	10.2%	
Weekdays	First day of the week	50	15.2%	28	14.5%	22	16.1%	0.571
	Weekdays	203	61.5%	116	60.1%	87	63.5%	
	Weekend	77	23.3%	49	25.4%	28	20.4%	

Abbreviations: ED, emergency department; ACS, acute coronary syndrome; MD, medical doctor; IQR, interquartile range.

**Table 5 diagnostics-14-02733-t005:** Distribution of TIMI and HEART scores in patients discharged versus hospitalized in the ED.

	Total		Discharge *n* = 193		Hospitalization *n* = 137		
	N	(%)	N	(%)	N	(%)	P
TIMI Score 0	110	33.3%	96	49.7%	14	10.2%	<0.001
1–3	195	59.1%	91	47.2%	104	75.9%	
4+	25	7.6%	6	3.1%	19	13.9%	
HEART Score ≤3	166	50.3%	135	69.9%	31	22.6%	<0.001
4–6	138	41.8%	53	27.5%	85	62%	
7+	26	7.9%	5	2.6%	21	15.3%	

Abbreviations: TIMI, Thrombolysis in Myocardial Infarction; HEART, history, ECG, age, risk factors, and troponin; P, probability value.

**Table 6 diagnostics-14-02733-t006:** Logistic regression analysis of risk factors for hospitalization in patients with chest pain.

Risk Factors for Hospitalization	OR (95%CI)	P
Primary residence nursing home	0.387 (0.198–0.759)	0.006
Nurse shift 23:00–07:00		0.047
15:00–23:00	0.494 (0.259–0.942)	0.032
07:00–15:00	0.440 (0.174–1.115)	0.083
s/p CVA TIA	3.188 (0.844–12.039)	0.087
Hyperlipidemia	3.514 (1.968–6.275)	<0.001
ECG changes without ischemic changes		0.027
with ischemic changes	7.992 (1.555–41.073)	0.013
Dysrhythmias without ischemic changes	0.679 (0.274–1.686)	0.405
Diaphoresis	7.043 (1.939–25.581)	0.003
Dyspnea	3.641 (1.828–7.252)	<0.001
Troponin TIMI	3.820 (1.245–11.715)	0.019

Abbreviations: s/p CVA TIA, status after cerebrovascular accident/transient ischemic attack; TIMI, Thrombolysis in Myocardial Infarction; ECG, electrocardiogram; P, probability value.

**Table 7 diagnostics-14-02733-t007:** Logistic regression analysis of risk factors for hospitalization including factors significant in the univariate analysis along with relevant clinical characteristics (see Appendix A).

Risk Factors for Hospitalization	OR (95%CI)	P
Primary residence—nursing home	0.381 (0.201–0.722)	0.003
Hyperlipemia	3.246 (1.794–5.876)	<0.001
Hypertension	1.959 (1.055–3.640)	0.033
Fatigue	1.885 (0.953–3.730)	0.069
Diaphoresis	8.525 (2.562–28.364)	<0.001
Dyspnea	2.897 (1.538–5.455)	<0.001

Abbreviations: OR, odds ratio; CI, confidence interval; P, probability value.

## Data Availability

All data generated or analyzed during this study are included in this article. Further inquiries can be directed to the corresponding author upon reasonable request.

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
