# Peer review of "Predictive Factors and Risk Assessment for Hospitalization in Chest Pain Patients Admitted to the Emergency Department"

_diagnostics, 2024, doi:10.3390/diagnostics14232733_

Round 1
Reviewer 1 Report
Comments and Suggestions for Authors
Dear Author,
Thank you for the detailed and comprehensive presentation of this interesting work.
The title is thorough and reflects the content well. The introduction provides sufficient background information.
The methods section requires additional details, including an explanation for using data from 2019 rather than more recent data.
Overall, the results are presented in tables and figures; however, please consider reducing the number of images and tables.
The discussion could be further expanded and analyzed in light of existing studies.
The conclusion should be expanded to better reflect the findings of the current study.
The references are adequate and relevant, but they need to be formatted according to the journal's style.
Author Response
We thank the editor and reviewers for their constructive comments and suggestions. All of the Editors' and reviewers’ recommendations have been implemented in the revised manuscript. Below are our point-by-point responses to the reviewers’ comments.
Reviewer 1
Dear Author,
Point 1: Thank you for the detailed and comprehensive presentation of this interesting work. The title is thorough and reflects the content well. The introduction provides sufficient background information. The methods section requires additional details, including an explanation for using data from 2019 rather than more recent data.
Response 1: We have addressed this by adding an explanation in the methods section (lines 78-86) clarifying the rationale for using 2019 data. Please see the revised text for details.
Point 2: Overall, the results are presented in tables and figures; however, please consider reducing the number of images and tables.
Response 2: We agree with the reviewer’s suggestion and have removed the flowchart, thereby reducing redundancy in the presentation of results.
Point 3: The discussion could be further expanded and analyzed in light of existing studies.
Response 3: We have expanded the discussion to include a more in-depth analysis of our findings in the context of existing literature. Specific additions have been made in lines 325-331, 349-354, 359-366, 373-391 and 405- 421. These additions provide a broader perspective and enhance the scientific rigor of the discussion.
Point 4: The conclusion should be expanded to better reflect the findings of the current study.
Response 4: The conclusion has been expanded to provide a more comprehensive summary of the findings and their implications. Please see lines 465-468 for the revised conclusion.
Point 5: The references are adequate and relevant, but they need to be formatted according to the journal's style.
Response 5: The references have been revised to conform to the journal's formatting requirements.
Reviewer 2 Report
Comments and Suggestions for Authors
I really appreciate the request to evaluate the paper titled "Predictive Factors and Risk Assessment for Hospitalization in Chest Pain Patients Admitted to the Emergency Department." This work is intriguing; nonetheless, several aspects need improvement. Firstly, the overlapping graphics hinder complete visibility, which I request the writers to rectify in the next version. This applies to fragments: Figure 1. (line 257-261) or line 301-303. The abstract is entirely accurate and adequate. I request a slight expansion of the introduction and the inclusion of information regarding biomarkers relevant to the discussed issue, as this would significantly enhance and underscore the potential for advancements in both diagnostics and prognostics related to the topic at hand. The methods used are well-detailed and accurate, as are the findings, which align with the conclusions. The discourse may be broadened to include novel methodologies that will facilitate the advancement of this domain, particularly when the writers address a comprehensive topic. The integration of existing scales with novel biomarkers or correlations, such as NLR or PLR, significantly enhances the discourse, supported by substantial data and recent meta-analyses on these subjects. It would be beneficial to relocate the limitations of the thesis to the closing portion of the discussion, as this would enhance coherence, particularly before the conclusions of the thesis. Nonetheless, the work is commendable and engaging; I commend the writers on an excellent manuscript, requiring just minor revisions.
Author Response
We thank the editor and reviewers for their constructive comments and suggestions. All of the Editors' and reviewers’ recommendations have been implemented in the revised manuscript. Below are our point-by-point responses to the reviewers’ comments.
Reviewer 2
I really appreciate the request to evaluate the paper titled "Predictive Factors and Risk Assessment for Hospitalization in Chest Pain Patients Admitted to the Emergency Department." This work is intriguing; nonetheless, several aspects need improvement.
Point 1: Firstly, the overlapping graphics hinder complete visibility, which I request the writers to rectify in the next version. This applies to fragments: Figure 1. (line 257-261) or line 301-303.
Response 1: The overlapping graphics, which previously hindered complete visibility, have been rectified in this version. Please see lines 258-264.
The abstract is entirely accurate and adequate.
Point 2: I request a slight expansion of the introduction and the inclusion of information regarding biomarkers relevant to the discussed issue, as this would significantly enhance and underscore the potential for advancements in both diagnostics and prognostics related to the topic at hand.
Response 2: We have expanded the introduction to include information on biomarkers relevant to the topic. Specific additions have been made in lines 60-64.
Point 3: The methods used are well-detailed and accurate, as are the findings, which align with the conclusions. The discourse may be broadened to include novel methodologies that will facilitate the advancement of this domain, particularly when the writers address a comprehensive topic.
Response 3: We have expanded the discussion to include an analysis of novel methodologies and their potential to advance this domain. Specific additions have been made in lines 316-322, 340-345, 350-357, 367-382, and 396- 412, ensuring a more comprehensive and forward-looking discussion.
Point 4: The integration of existing scales with novel biomarkers or correlations, such as NLR or PLR, significantly enhances the discourse, supported by substantial data and recent meta-analyses on these subjects. It would be beneficial to relocate the limitations of the thesis to the closing portion of the discussion, as this would enhance coherence, particularly before the conclusions of the thesis.
Response 4:
We have added data on NLR and PLR, in the results section, and we have expanded the discussion on these inflammatory factors. Specific additions have been made in lines 209, 256-257, 363-379
We have relocated the limitations section to the closing portion of the discussion for improved coherence and logical flow. The updated limitations section is now presented in lines 448-452.
Nonetheless, the work is commendable and engaging; I commend the writers on an excellent manuscript, requiring just minor revisions.
Reviewer 3 Report
Comments and Suggestions for Authors
The study identifies critical factors affecting the likelihood of hospitalization of patients with chest pain admitted to the emergency department.
I have no substantive comments. Some editing of the text is required.
Authors need to check the text for typos (punctuation) and the use of abbreviations. Line 58 - 'ER' instead of 'ED'?
Figures in the text and tables should not be repeated.
The caption to Table 2 contains abbreviations that are not in the table itself.
Tables and figures should be formatted! The numbering of figures is not correct.
Table 3. The value of the white blood cell (WBC) count is better written as 1000/ul. Remove the second line. Check the units of measurement. It is common to give the mean +/- SD, not mean + SD.
The p-value should not be deciphered in the table captions.
The authors write ‘All data collection procedures were conducted in accordance with the guidelines of the local ethics (Helsinki) committee’. This phrase is unclear. There is a local committee that works following the Helsinki Declaration.
Author Response
We thank the editor and reviewers for their constructive comments and suggestions. All of the Editors' and reviewers’ recommendations have been implemented in the revised manuscript. Below are our point-by-point responses to the reviewers’ comments.
Reviewer 3
Point 1: The study identifies critical factors affecting the likelihood of hospitalization of patients with chest pain admitted to the emergency department. I have no substantive comments. Some editing of the text is required. Authors need to check the text for typos (punctuation) and the use of abbreviations. Line 58 - 'ER' instead of 'ED'?
Response 1: We have thoroughly reviewed the manuscript for typos, punctuation errors, and the consistent use of abbreviations.
Point 2: Figures in the text and tables should not be repeated.
Response 2: To address this, we have deleted Chart 1 and removed redundant explanations in the text for Table 1 and Table 2. This ensures that the figures and tables are presented concisely without unnecessary repetition.
Point 3: The caption to Table 2 contains abbreviations that are not in the table itself.
Response 3: This issue has been corrected by ensuring that all abbreviations in the caption to Table 2 are either included in the table or clarified within the caption itself.
Point 4: Tables and figures should be formatted! The numbering of figures is not correct.
Response 4: The formatting of tables and figures has been reviewed and corrected. The numbering of figures has also been updated to ensure accuracy and consistency throughout the manuscript.
Point 5: Table 3. The value of the white blood cell (WBC) count is better written as 1000/ul. Remove the second line. Check the units of measurement. It is common to give the mean +/- SD, not mean + SD.
Response 5: The WBC value has been corrected to 1000/μl, and the second line has been removed. Additionally, the units of measurement have been reviewed and corrected, and the mean values are now presented as mean ± SD.
Point 6: The authors write ‘All data collection procedures were conducted in accordance with the guidelines of the local ethics (Helsinki) committee’. This phrase is unclear. There is a local committee that works following the Helsinki Declaration.
Response 6: We have clarified this statement to accurately reflect that the local ethics committee operates in accordance with the Helsinki Declaration. The corrected version can be found in lines 78-86.
Reviewer 4 Report
Comments and Suggestions for Authors
I do not fully understand the purpose of this study. If the authors aimed to propose an algorithm for ER physicians to follow when deciding on hospitalizing patients with chest pain, this algorithm should have been explicitly outlined. Moreover, I believe it would have been helpful to clearly rank the TIMI and HEART scores for risk stratification and decision-making regarding hospitalization. I think the data collected by the authors allow for at least somewhat more definitive conclusions to be drawn. The fact that the HEART score has a higher prognostic value than the TIMI score does not surprise me personally, as the HEART score encompasses the most significant risks of coronary disorders.
Some comments on the article's text, both minor and more serious:
1. In Table 3, it is surprising to see no difference in creatinine levels between the groups, despite a significant difference in glomerular filtration rate (GFR). How was GFR assessed in the ER if not based on creatinine levels?
2. In Table 4, it would be helpful to specify which arrhythmias led to a refusal of hospitalization. Does this concern only extrasystole, or were patients with more serious rhythm disturbances also denied hospitalization?
3. Section 3.3. It would be valuable to see which pathology (or symptoms) most often led to readmission and how this relates to the HEART and TIMI scores.
Minor points:
1. Line 121: should be "HEART" in capitals.
2. Figure 1: should be "refeRRal."
3. Table 5: "TII Score 0" should probably be "TIMI."
4. Table 7 and other parts of the text (e.g., line 286): "hyperlipemia" should likely be "hyperlipidemia."
Author Response
We thank the editor and reviewers for their constructive comments and suggestions. All of the Editors' and reviewers’ recommendations have been implemented in the revised manuscript. Below are our point-by-point responses to the reviewers’ comments.
Reviewer 4
Point 1: I do not fully understand the purpose of this study. If the authors aimed to propose an algorithm for ER physicians to follow when deciding on hospitalizing patients with chest pain, this algorithm should have been explicitly outlined. Moreover, I believe it would have been helpful to clearly rank the TIMI and HEART scores for risk stratification and decision-making regarding hospitalization.
Response 1: The aim of the study, clarified in lines 316-322, The study aimed to identify predictive factors and assess the risk of hospitalization for patients presenting with chest pain in the emergency department (ED). By analyzing demographic data, clinical symptoms, laboratory findings, and risk scores, the study sought to differentiate between patients who required hospitalization and those who could be safely discharged. Additionally, the study evaluated the predictive value of established scoring systems, specifically the HEART and TIMI scores, to enhance clinical decision-making and optimize resource allocation in ED settings.
Point 2: I think the data collected by the authors allow for at least somewhat more definitive conclusions to be drawn. The fact that the HEART score has a higher prognostic value than the TIMI score does not surprise me personally, as the HEART score encompasses the most significant risks of coronary disorders.
Response 2: More definitive conclusions have been drawn and expanded in lines 396-412, emphasizing the HEART score's superior prognostic value due to its incorporation of key coronary risk factors.
Some comments on the article's text, both minor and more serious:
- Point 3: In Table 3, it is surprising to see no difference in creatinine levels between the groups, despite a significant difference in glomerular filtration rate (GFR). How was GFR assessed in the ER if not based on creatinine levels?
Response 3: Estimated GFR (eGFR) is calculated in the emergency department using automated formulas. While GFR is influenced by creatinine levels, it is also affected by age. The formula incorporates creatinine, age, and sex. Given the significant age differences between the groups, it is unsurprising that eGFR values differed from creatinine levels alone.
- Point 4: In Table 4, it would be helpful to specify which arrhythmias led to a refusal of hospitalization. Does this concern only extrasystole, or were patients with more serious rhythm disturbances also denied hospitalization?
Response 4: Most arrhythmias discharged from the emergency department are benign and do not require hospital evaluation. However, there are cases involving more significant arrhythmias, such as atrial fibrillation in patients with paroxysmal AF (PAF) or supraventricular tachycardia (SVT), which are managed within the emergency department. Given the low number of cases with VT (3 cases), it was decided not to include them as a separate category in the analysis. Added to methods in lines 125-129.
- Point 5: Section 3.3. It would be valuable to see which pathology (or symptoms) most often led to readmission and how this relates to the HEART and TIMI scores.
Response 5: This is indeed an important question. However, in our study, the number of readmissions within 24 to 72 hours post-discharge was relatively small, limiting robust statistical analysis. Future studies with larger cohorts are required to validate these findings and explore the association between specific pathologies and readmission risk.
Minor points:
- Line 121: should be "HEART" in capitals. Corrected
- Figure 1: should be "refeRRal." Corrected
- Table 5: "TII Score 0" should probably be "TIMI." Corrected
- Table 7 and other parts of the text (e.g., line 286): "hyperlipemia" should likely be "hyperlipidemia." Corrected